# Conductance quantization suppression in the quantum Hall regime

José M. Caridad [1], Stephen R. Power [1,2,3], Mikkel R. Lotz[1], Artsem A. Shylau[1], Joachim D. Thomsen[1], Lene Gammelgaard[1], Timothy J. Booth [1], Antti-Pekka Jauho[1] & Peter Bøggild [1]

Conductance quantization is the quintessential feature of electronic transport in non-interacting mesoscopic systems. This phenomenon is observed in quasi one-dimensional conductors at zero magnetic field B, and the formation of edge states at finite magnetic fields results in wider conductance plateaus within the quantum Hall regime. Electrostatic inter-actions can change this picture qualitatively. At finite B, screening mechanisms in narrow, gated ballistic conductors are predicted to give rise to an increase in conductance and a suppression of quantization due to the appearance of additional conduction channels. Despite being a universal effect, this regime has proven experimentally elusive because of difficulties in realizing one-dimensional systems with sufficiently hard-walled, disorder-free confinement. Here, we experimentally demonstrate the suppression of conductance quantization within the quantum Hall regime for graphene nanoconstrictions with low edge roughness. Our findings may have profound impact on fundamental studies of quantum transport in finite-size, two-dimensional crystals with low disorder.

[1] Center for Nanostructured Graphene (CNG), Department of Micro- and Nanotechnology, Technical University of Denmark, 2800 Kongens Lyngby, Denmark. [2] Catalan Institute of Nanoscience and Nanotechnology (ICN2), CSIC and The Barcelona Institute of Science and Technology, Campus UAB, Bellaterra, Barcelona 08193, Spain. [3] Universitat Autònoma de Barcelona, Bellaterra (Cerdanyola del Vallès) 08193, Spain. José M. Caridad and Stephen R. Power contributed equally to this work. Correspondence and requests for materials should be addressed to J.M.C. (email: jcar@nanotech.dtu.dk) or to P.Bøg. (email: peter.boggild@nanotech.dtu.dk)

At zero magnetic field **B**, conductance quantization arises due to the formation of transverse subbands in confined, quasi one-dimensional (1D) systems such as quantum point contacts (QPC) or quantum wires[1,2]. As **B** increases, the system gradually enters the quantum Hall (QH) regime, where propagating modes evolve from magnetoelectric subbands interacting with both edges, to chiral edge states surrounding an incompressible, gapped bulk[1,2]. Within a one-electron picture, both propagation states lead to a quantized two-terminal conductance given by $G = Ne^2/h$ (here, $e$ is the electron charge, $h$ the Plank constant and $N$ the number of conducting modes at the Fermi level)[1,2]. The situation changes when taking into account Coulomb interactions[3–13] between injected carriers and/or their coupling to an external gate. For example, the conductance of a 1D channel with repulsive electron–electron interactions vanishes in the presence of any scattering potential at $|\mathbf{B}| = 0$ T ref[3]. Furthermore, the observation of the so-called 0.7 anomaly[12] or the 0.25 feature[13] in the conductance quantization of QPCs at $|\mathbf{B}| = 0$ T are also signatures of electron–electron interactions. In a perpendicular **B**, the interplay between screening mechanisms and the Hall potential causes a reconstruction of the edge states into alternating conductive (compressible) and insulating (incompressible) regions no longer strictly linked to the topology of the conductor[4–8]. Compressible zones are characterized by partially filled Landau levels (LLs) pinned at the Fermi energy with a variable electron concentration. Conversely, incompressible regions (strips) consist of fully occupied LLs and display the typical insulating behavior of a QH state[4–8].

We focus on the ballistic conductance of gated quasi-1D systems, where screening theories predict conductance quantization suppression (CQS) in the QH regime[4–9]. This universal transport regime should occur in narrow, ballistic systems confined by hard-wall potentials[4,9,11], where a large accumulation of charge carriers near sharp edges and a pronounced inner depletion inhibits the formation of stable incompressible strips[4–9]. Although both interactions between carriers and their coupling to the external gate can affect conductance[6,9], it is the electrostatic screening of the gate potential which is the main contributing mechanism[4,5,10] to the CQS effect.

To date, the experimental realization of such narrow, disorder-free, sharp-edged devices has been inherently difficult[1,2,14–24]. Commonly studied QPCs in two-dimensional (2D) electron gases have soft-confining potentials because the gates and dopant layers are far away from the actual carrier layer[1,2]. Graphene, on the other hand, provides extraordinary opportunities to examine the physics of the QH effect[16]. First, it exhibits a natural hard-wall confinement at its borders. Furthermore, the distance to the gate can be arbitrarily selected since electrons in strict 2D materials reside right at the surface. Both features enable the possibility of designing specific device geometries which are (heavily) dominated by screening effects. An example of such a geometry is a narrow graphene strip with a width comparable or smaller than the thickness of the dielectric spacer[9,10]. Indeed, CQS in a perpendicular magnetic field has been predicted to occur in gated graphene nanoribbons[9,11]. In these systems, the suppression of conductance quantization is related solely to the simultaneous existence of compressible strips in the center of the ribbon and the appearance of additional counter-propagating states[9,11]. Nevertheless, experiments conducted with different types of narrow, high-quality graphene devices have so far not confirmed these predictions[17–24].

In more detail, the magnetoconductance of ballistic graphene constrictions remains quantized when increasing **B**[17,18], similar to gate-defined, soft-potential, narrow ballistic graphene channels[19]. These discrepancies between experiments and theoretical predictions motivate us to investigate devices which have been designed to meet the required theoretically predicted conditions for CQS;[9,11] specifically, a device geometry able to produce a large charge density gradient across the nanostructure, a narrow ballistic channel, and low edge disorder.

By addressing these factors, we experimentally demonstrate the suppression of conductance quantization within the QH regime for graphene nanoconstrictions with low edge roughness. Our findings are a strong experimental confirmation that the single-electron picture is inadequate for describing the transport behavior of finite-size, two-dimensional crystals with low disorder.

## Results

**Design of narrow devices free of incompressible strips.** According to QH theories[4–8], incompressible strips must be wider than the magnetic length $l_B$ to be stable. For a given LL with level index $k$, the minimum charge carrier density gradient across a graphene nanostructure, which prevents the formation of a stable incompressible strip, is (Methods)

$$\left. \frac{dn_{el}(x)}{dx} \right|_{min, x \in \left[ \frac{-W}{2}, \frac{+W}{2} \right]} = \nabla n_{el}|_{min} = \frac{\varepsilon v_F}{\pi^2} \left( \frac{|k|}{\hbar e} \right)^{1/2} (2|\mathbf{B}|)^{3/2}, \quad (1)$$

where $\varepsilon$ is the permittivity of the dielectric and $v_F \sim 10^6$ ms$^{-1}$ the Fermi velocity in graphene.

Figure 1a shows $\nabla n_{el}|_{min}$ as a function of $|\mathbf{B}|$ and $k$, normalized by the average density $n_{avg} = 10^{16}$ m$^{-2}$ ($\overline{\nabla n_{el}}|_{min}$), using SiO$_2$ as dielectric material. Values of $|\mathbf{B}|$ of 0–10 T, $k$ of 0, 1, 2, and $n_{avg}$ are experimentally accessible in our study. For $|\mathbf{B}| \leq 10$ T and $k \leq 2$, an estimated threshold of $\overline{\nabla n_{el}}|_{min} = C \sim 10^7$ m$^{-1}$ prevents incompressible strips from forming in graphene devices. Figure 1b, c show the simulated normalized electron density $\overline{n_{el}(x)}$ and $\overline{\nabla n_{el}(x)}$ across three quasi-1D systems, respectively.

Here, we consider two distinct geometries (ribbons and constrictions) with different widths $W$ and dielectric thicknesses $b$ to examine the stability condition (Eq. (1)). The ribbon geometry ($W = 50$ nm, $b = 300$ nm) used for the theoretical prediction of the CQS[9] (green curve) shows $\overline{\nabla n_{el}(x)} > C$ for distances $x > 0.13W$ across the device. This length is comparable to $l_B = 0.16W$ at $|\mathbf{B}| = 10$ T, preventing the appearance of stable incompressible strips. A similar situation occurs (blue curve) in slightly wider constrictions ($W = 100$ nm) on a dielectric with $b = 100$ nm. Notably, wider geometries have the added advantage of reducing the significance of edge disorder in experimental devices. Much wider constrictions with sizes close to samples reported in literature[17,18] ($W = 300$ nm, red curve) show $\overline{\nabla n_{el}(x)} \sim C$ at distances an order of magnitude larger than $l_B$ at $|\mathbf{B}| = 10$ T. This condition remains satisfied for smaller $|\mathbf{B}|$ and $k$, and so this geometry enables the formation of incompressible strips[4–8] and results in a quantized magnetoconductance[17,18].

**Fabrication of graphene nanoconstrictions.** Guided by these simulations, we fabricate (Fig. 1d) graphene nanoconstrictions with length $L = W \sim 100$ nm on $b = 100$ nm SiO$_2$ substrates (Methods). Our graphene flakes were exfoliated on hydrophobic SiO$_2$[25], resulting in mean free paths larger than $L, W$ ($l_{mfp} \sim 200$ nm at a temperature $T = 4$ K, Supplementary Figs. 1 and 2 and Supplementary Note 1). Figure 2 shows the magnetoconductance $G = G(V_g, |\mathbf{B}|)$ of the two types of studied sample. Their geometry and fabrication steps are similar with the exception of the last etching step, which defines the edge disorder of the nanoconstriction[26]. While all our devices have a certain degree of edge disorder, Sample type 1 (Fig. 2a) was etched using reactive ion etching (RIE), which is known to produce less edge disorder than the oxygen plasma ashing[26] technique used in Sample type 2

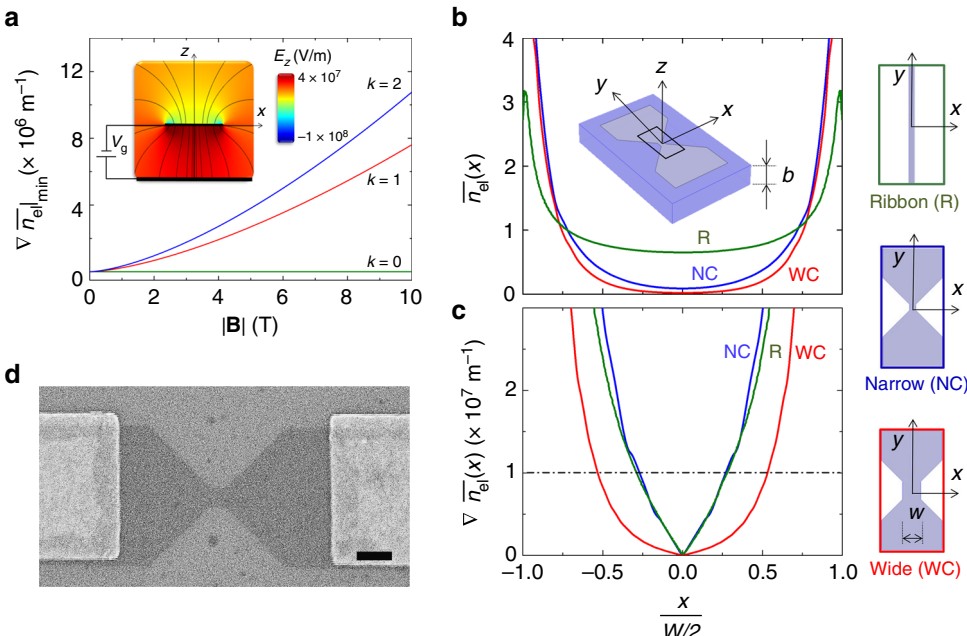

**Fig. 1** Electrostatic design and fabrication of graphene nanoconstrictions. **a** Minimum normalized carrier density gradient $\nabla \overline{n_{el}}|_{min}$ (Eq. (1)) required for a graphene nanostructure to prevent incompressible strips (LLs $k = 0, 1, 2$; $|\mathbf{B}| = 0$–10 T and $n_{avg} = 1 \times 10^{16}$ m$^{-2}$). Inset: Inhomogeneous electric field $E_z$ close to narrow graphene nanostructures, separated by a thin SiO$_2$ layer of thickness $b$ from a back-gate electrode ($W = b$). **b** Normalized carrier density $\overline{n_{el}(x)}$ and **c** carrier density gradient $\nabla \overline{n_{el}(x)}$ across three representative nanostructures: nanoribbons (R, green, $b = 300$ nm and $W = 50$ nm); narrow constrictions (NC, blue $b = 100$ nm and $W = 100$ nm); and wide constrictions (WC, red, $b = 300$ nm and $W = 300$ nm). Dash-dot line indicates the threshold C (main text). Inset in panel **b** shows a schematic of one of the simulated devices. **d** Scanning electron micrograph of a nanoconstriction device. Scale bar is 200 nm

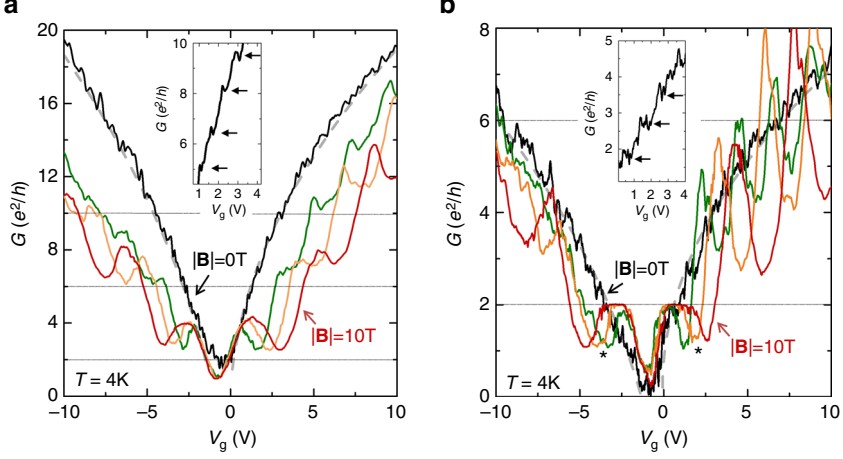

**Fig. 2** CQS effect. **a** Conductance $G$ vs gate voltage $V_g$ in a nanoconstriction (Sample type 1) with low edge roughness at different magnetic fields $|\mathbf{B}| = 0$ T (black), 6 T (green), 8 T (orange), and 10 T (red). Gray-dashed lines are fit to the data, $G \propto \left(\Delta V_g\right)^{1/2}$. The extracted contact resistance $R_c$ in this device is 410 Ω. Inset shows periodic conductance modulations with step heights $\Delta G \sim 2e^2/h$. **b** $G(V_g)$ in a nanoconstriction with larger edge disorder (Sample type 2) at different magnetic fields $|\mathbf{B}| = 0$ T (black), 6 T (green), 8 T (orange), and 10 T (red). Gray-dashed lines are fit to the data, $G \propto \left(\Delta V_g\right)^{1/2}$. The extracted $R_c$ in this device is 518 Ω. Inset shows periodic conductance modulations with step heights $\Delta G \sim e^2/h$

(Fig. 2b). Specifically, we achieve an edge roughness ≤1 nm in Sample type 1 (Supplementary Fig. 3 and Supplementary Note 2). This value is comparable to values obtained in nanoribbons with extremely low edge roughness fabricated by unzipping carbon nanotubes[20].

**Experimental observation of the CQS effect**. At zero **B**, both types of sample show $G \propto \left(\Delta V_g\right)^{1/2}$, characteristic of transport limited by boundary scattering[18,27]. However, conductance values in Sample type 1 are three times larger than those for Sample type 2. This specific behavior has previously been attributed to differences in edge disorder[26]. Moreover, the conductance of these samples shows periodic modulations (arrows in the insets), a clear indication of size quantization[18]. These modulations are significant in Sample type 1, with step heights $\Delta G$ up to ~ $2e^2/h$. Further analysis at $|\mathbf{B}| = 0$ T can be found in the Supplementary Information (Supplementary Figs. 4–7 and Supplementary Note 3).

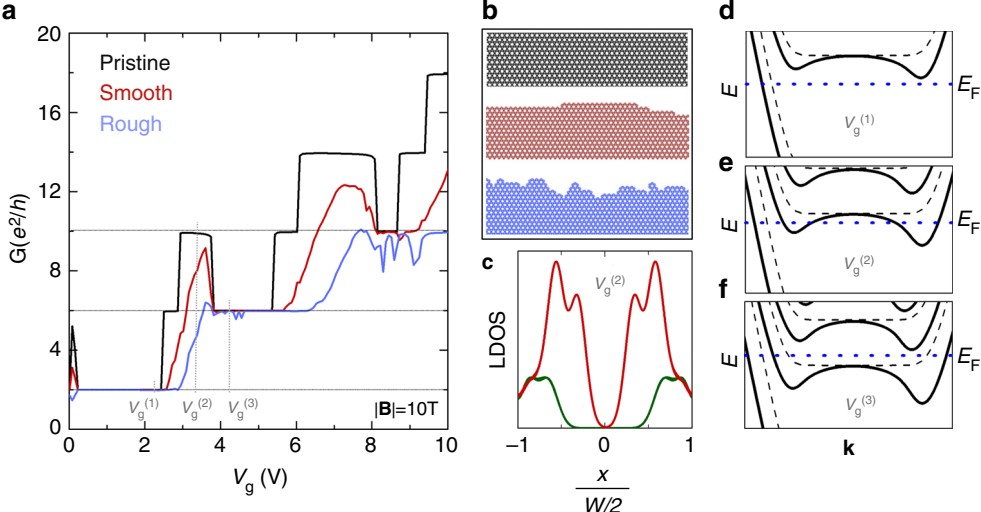

**Fig. 3** Tight-binding simulation of CQS. **a** Simulated conductance, including non-uniform gating effects, at |**B**| = 10 T for a pristine nanoribbon (black), and ribbons with smooth (red) and rough (blue) edges. The CQS peaks are only fully suppressed by the stronger disorder (see Methods). **b** Schematics of the different edges considered in **a**. **c** Local density of states (LDOS) across a pristine ribbon at a gate voltage $V_g^{(2)}$ that displays a CQS peak. Non-uniform gating (red) introduces states in the ribbon bulk, which are not present for uniform gating (green). **d–f** Band structure near the Fermi level $E_F$ for three gate voltages ($V_g^{(1)}$, $V_g^{(2)}$, $V_g^{(3)}$), marked in panel **a** near a CQS peak, respectively. Non-uniform gating introduces a clear distortion of the band structure relative to the uniform case (dashed), which opens new conductance channels

For |**B**| ≠ 0 T, the two sample types exhibit distinctly different behavior. The conductance is not quantized for any of the three shown LLs for Sample type 1 (smooth edges), dramatically differing from the single-electron picture. Particularly, when increasing the gate voltage $V_g$, $G$ shows a peak whose value is larger than the expected quantization plateau and cannot be explained by accounting for geometrical corrections in spatially uniform and homogeneous conductors[28] (Supplementary Fig. 8 and Supplementary Note 4). These are signature features of CQS[9], and are predicted to disappear with increasing disorder[11].

This is confirmed for Sample type 2 (larger edge disorder), which exhibits a quantized $G$ at $k = 0$ (Fig. 2b). Furthermore, $G$ at $k = 0$ in Sample type 2 presents a dip after the plateau (marked with '*'), in agreement with the well-known, geometrical effects for homogeneous devices with $L > W$ [28] (Supplementary Fig. 8 and Supplementary Note 4). Disorder is similarly responsible for LLs $k = 1$, 2 in Sample type 2 showing $G$ values lower than the expected quantization values. These trends are confirmed in further devices and analysis (Supplementary Figs. 9, 10, and 15 and Supplementary Notes 5 and 6). Importantly, the extreme sensitivity to device electrostatics and edge disorder demonstrated here explains the absence of the CQS phenomenon (Supplementary Fig. 11 and Supplementary Note 6) in graphene devices previously reported in literature[17,18,20–24] (Supplementary Table 1 and Supplementary Note 7). These two effects are related: the presence of edge roughness leads to a flatter $\overline{n_{el}(x)}$ even within an electrostatic approach (Supplementary Fig. 12 and Supplementary Note 8).

**Theoretical analysis of the CQS effect.** The interpretation given above is supported by tight-binding calculations, where the inhomogeneous electrostatic potential across the device is introduced using the analytical model proposed by Silvestrov and Efetov[10] (Methods). This potential corresponds closely to those generated by more complex models, such as self-consistent solutions within the Hartree approximation[9,27,29]. Although such approaches can account for both Coulomb

interactions between injected carriers and their coupling to the external gate, the electrostatic screening of the gate potential is the primary factor determining the charge density distribution in these systems[30].

Figure 3a shows the calculated conductance with pristine edges, and with smooth and rough edge disorder. In pristine systems, the screening potential gives rise to additional conduction channels, causing a larger, quantized conductance to appear near the onset of the expected QH plateaus.

Unlike QH edge states, these conductance peaks are associated with new states with finite weight over a large portion of the ribbon's width (Fig. 3c), which emerge due to a bending of the previously dispersionless LLs by the spatially varying gate potential (Fig. 3d–f). This is equivalent to the formation of compressible strips in the system[9]. These new dispersive states support propagation in both directions and unlike QH states, are susceptible to backscattering due to the overlap between forward and backward propagating states[9] and disorder[9,11]. Therefore the channels lose their exact quantization as edge disorder is increased, forming peak-like conductance features at low edge disorder levels, before being completely suppressed by stronger scattering (Fig. 3a). Our simulations confirm that CQS is still appreciable at low edge roughness (Fig. 3b), similar to that present in our Sample type 1 (≤1 nm), and vanishes for stronger edge disorder, in agreement with our observations on Sample type 2.

## Discussion

We have demonstrated the suppression of conductance quantization in the QH regime due to electrostatic interactions in gated graphene nanoconstrictions with low edge roughness. Although demonstrated here in graphene, this is a universal phenomenon[4,9–11,27] occurring in ballistic, narrow conducting systems exhibiting hard-wall potential confinement such as semiconducting and metallic 2D crystals or cleaved-edged overgrown quantum wires[31]. In a wider perspective, our study demonstrates radical disruptions of the conduction properties of atomically thin materials subject to inhomogeneous electron density distributions, emphasizing the critical relevance of device geometries and

processing methods when studying interacting-electron transport physics in nanoscaled devices. Our findings have particular relevance for quantum transport and information studies[15,16], the production of resistance standards,[15,16] and plasmonics[32].

## Methods

**Graphene nanoconstrictions free of incompressible strips.** The competition between the Hall and screened potentials determines the stability of incompressible strips in a perpendicular magnetic field[4-8]. The condition for a stable strip of width $a_k$ with level index $k$ requires $a_k > l_B$, where $l_B = \left( \hbar e^{-1} |\mathbf{B}|^{-1} \right)^{1/2}$ is the magnetic length. Although the stability/collapse of an incompressible strip is a direct finding of a self-consistent calculation, a rough estimate of the stability condition can be done from electrostatic calculations by Chklovskii et al.[4,7]. According to this theory, $a_k$ is estimated by the equation:

$$a_k = \sqrt{\frac{2\varepsilon E_k}{\pi^2 e^2 \left( \frac{dn_{el}(x)}{dx} \Big|_k \right)}}, \qquad (2)$$

where $n_{el}(x)$ is the electron density across the device at $|\mathbf{B}| = 0$ T, $\varepsilon$ is the dielectric constant of the insulating material, $\frac{dn_{el}(x)}{dx} \Big|_k$ is the charge density gradient evaluated at the center of the $k$th incompressible strip, and $E_k$ is the Landau spectrum. In the case of spin-degenerate graphene, we get $E_k = \left( 2e\hbar v_F^2 |\mathbf{B}| |k| \right)^{1/2}$[16]. Thus, for the graphene nanodevice to be free of incompressible strips, the charge carrier density gradient $\frac{dn_{el}(x)}{dx}$ across the nanostructure has to obey the following inequality:

$$\frac{dn_{el}(x)}{dx} \geq \frac{\varepsilon v_F}{\pi^2} \left( \frac{|k|}{\hbar e} \right)^{1/2} (2|\mathbf{B}|)^{3/2}, \qquad (3)$$

where the corresponding equality is Eq. (1) in the main text.

**Electrostatic simulations.** Spatial carrier density profiles across graphene devices $n_{el}(x)$ can be calculated[33] using the expression $n_{el}(x) = \frac{\varepsilon}{e} E_z(x)$, where $E_z(x)$ is the perpendicular electric field component in the corresponding gated devices at $y = 0$ at a distance $z = 0.5$ nm above the flake and $\varepsilon = 3.9\varepsilon_0$ is the permittivity of the $SiO_2$. $E_z(x)$ can be obtained for any geometry by solving the Poisson equation in the device using a finite-element method[33] solver (Fig. 1a, inset).

The carrier density profile normalized with respect to the average electron density across the constriction $n_{avg}$ is then given by $\frac{n_{el}(x)}{n_{avg}} = \frac{E_z(x)}{E_{avg}}$, where

$$E_{avg} = \frac{\int_{-W/2}^{W/2} E_z(x) dx}{W} \qquad (4)$$

is a fictitious electric field across the constriction which would generate $n_{avg}$. We note how in the case of nanoribbon geometries (Fig. 1b, green), the numerically calculated $\frac{n_{el}(x)}{n_{avg}}$ agrees excellently with the analytical expression obtained in ref. [10] (Supplementary Fig. 13).

**Fabrication of graphene nanoconstrictions.** We fabricate devices with field-effect mobility $\mu \sim 20{,}000$ cm$^2$ V$^{-1}$ s$^{-1}$ (estimated mean free paths $l_{mgp} \sim 200$ nm), achieved by the mechanical exfoliation of graphene on hydrophobic[25] Si/SiO$_2$ substrates (SiO$_2$ thickness $b = 100$ nm) and contact resistance $R_c$ below 600 Ω. To test these initial device parameters ($\mu$, $R_c$), we first shape, contact and measure the magnetotransport properties of rectangular two-terminal devices with a width of ~1 μm (Supplementary Fig. 1). This is a common procedure undertaken to assess the graphene quality[24] before patterning the actual nanoconstriction devices. We contact these devices by evaporating Ti (5 nm) and Au (30 nm) at low pressure (<5 × 10$^{-7}$ mbar). The subsequent definition of the nanoconstrictions is done via electron beam lithography using polymethyl-methacrylate developed at −5 °C in a 1:3 IPA:H$_2$O solution.

The edge quality in our constrictions is defined with two complementary etching processes: oxygen plasma ashing and RIE[26]. Devices with a higher amount of edge disorder (Sample type 2) are defined by plasma ashing, which, despite being known to introduce instabilities and localized states in graphene nanodevices, is widely used to shape graphene nanostructures[24]. In contrast, devices with a much lower amount of edge disorder (Sample type 1) were produced by RIE[26] (power ~40 W, argon 40 sccm, oxygen 5 sccm). We achieve an edge roughness ≤1 nm with the RIE etching procedure, as demonstrated in the transmission electron micrograph shown in the Supplementary Fig. 3.

Prior to measuring their electrical properties, we dip our devices for 18 h in a pure hexamethyldisilazane solution to reduce the effect of environmental contaminants that may have been adsorbed on the basal plane of graphene or at the edges during the processing steps[34]. After these 18 h, the devices are dipped for 5 s in acetone, 5 s in IPA, and then dried with nitrogen.

**Electrical measurements.** Our measurements were done in an Oxford Instrument Teslatron PT cryostat. Measurements of differential conductance were performed using a Stanford SR830 lock-in amplifier with an excitation voltage of 80 μV at a frequency of 17.77 Hz.

**Tight-binding calculations.** In our simulations, we consider zigzag nanoribbons with similar dimensions to the experimentally measured constrictions ($L = W = 100$ nm) and different degrees of edge disorder (Supplementary Note 9). Additionally, device leads are formed by semi-infinite pristine nanoribbons.

The electronic structure is described by a single π-orbital third-nearest-neighbor tight-binding Hamiltonian

$$H = \sum_{<ij>} t_{ij}(\mathbf{B}) \hat{c}_i^\dagger \hat{c}_j, \qquad (5)$$

where $\hat{c}_i^\dagger$ ($\hat{c}_i$) are the creation and annihilation operators associated with lattice site $i$. The hopping parameters $t_{ij}$ take the values $t_1 = -2.7$ eV, $t_2 = -0.2$ eV, and $t_3 = -0.18$ eV, respectively[35].

The effect of a magnetic field is included using the Peierls' phase approach. This involves introducing a field-dependent phase factor in the tight-binding hopping parameters

$$t_{ij}(\mathbf{B}) = t_{ij}(0) e^{\frac{2\pi i e}{\hbar} \Theta_{ij}}, \qquad (6)$$

where

$$\Theta_{ij} = \int_{r_i}^{r_j} \mathbf{A}(\mathbf{r}') d\mathbf{r}'. \qquad (7)$$

We choose the Landau gauge $\mathbf{A}_0 = |\mathbf{B}| x \hat{y}$ to maintain periodicity in the $y$-direction.

The conductance through the ribbon is evaluated in terms of the transmission

$$T(E) = T_r \left[ G^R(E) \Gamma_R(E) G^A(E) \Gamma_L(E) \right], \qquad (8)$$

where $G^R$ and $G^A$ are the retarded and advanced Green's functions respectively.

The effect of a gate voltage is introduced by fixing the Fermi energy and instead changing the onsite energy potentials according to

$$U(x) = -\hbar v_F \sqrt{\pi n_{el}(x)}. \qquad (9)$$

A uniform carrier density can be included using an infinite plane capacitor[25] $n_0$

$$n_{el}(x) = n_0 = \text{sgn}(V_g) \frac{\varepsilon V_g}{eb}, \qquad (10)$$

while non-uniform gating profiles can be approximated by the expression[10]

$$n_{el}(x) = \frac{n_{avg} W}{\pi \sqrt{(W/2)^2 - x^2}}, \qquad (11a)$$

or equivalently

$$n_{el}(x) = \frac{n_0 W}{2 \sqrt{(W/2)^2 - x^2}}. \qquad (11b)$$

Furthermore, in Fig. 3a we include a small shift (~0.2 V) to separate the charge neutrality and zero-gating points. This is necessary to observe the very narrow CQS peak for the LL0, which would otherwise coincide with zero gating, and thus a uniform potential. In experiments, additional sources of non-uniform charge density near the CNP can play a similar role. For example, a notable charge density accumulation can occur at edges due to dangling bonds and trapped charges[18,26], which gives rise to the stronger CQS observed for LL0 in our experiments (Supplementary Fig. 14 and Supplementary Note 10).

**Data availability.** The data that support the findings of this study are available from the corresponding authors on request.

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

## Acknowledgements

We acknowledge stimulating discussions with B. Terrés and thank E. Díez for the critical reading of the manuscript. This work was supported by the Danish National Research Foundation Center for Nanostructured Graphene, project DNRF103, the EU Seventh Framework Programme (FP7/2007-2013) under grant agreement number FP7-6040007 'GLADIATOR' and the EU H2020 Graphene Flagship Core 1, Grant Agreement No. 696656. The A.P. Møller and Chastine Mc-Kinney Møller Foundation is acknowledged for their contribution towards the establishment of the Center for Electron Nanoscopy at the Technical University of Denmark. S.R.P. acknowledges funding from the European Union's Horizon 2020 research and innovation programme under the Marie Skłodowska-Curie grant agreement No 665919. ICN2 is funded by the CERCA Programme/Generalitat de Catalunya and supported Severo Ochoa programme (MINECO, Grant. No. SEV-2013-0295).

## Author contributions

J.M.C. conceived the idea with valuable input from S.R.P. J.M.C. carried out the electrostatic design of the devices, measured their electrical properties and interpreted the data. J.M.C., M.R.L. and L.G. fabricated the devices. S.R.P. undertook the tight-binding simulations. J.D.T. and T.B. realized the TEM images. J.M.C., S.R.P. and A.S. carried out the data analysis. J.M.C., S.R.P., A.-P.J. and P.B. wrote the manuscript, with comments from all the authors.

## Additional information

**Competing interests:** The authors declare no competing financial interests.

