## [Peer Review File · Nature Communications]

Reviewers' comments:

Reviewer #1 (Remarks to the Author):

The manuscript "Conductance quantization suppression in the quantum Hall regime" by Bøggild et.al is on an interesting topic of interplay between space and B-field quantization in 2D materials.

The manuscript seems to be free from obvious errors, and can be published with suggested minor revisions, as below.

1. Perhaps main reservation Referee had regarding Authors interpretation of term "electron interactions". Although broadly defined, this term has specific meaning of Coulomb e-e correlation, especially when using in contrast to "non-interacting electrons".

What is described in the paper is much simpler effect (not necessarily this is a drawback) -- the electrostatic screening of the gate potential by non-uniform distribution of charge carriers, resulting from both E- and B-fields *and* involving disorder of the edge.

Current manuscript is a big mess in this sense. Word "interaction" is overused and makes unprepared reader completely confused.

2. Referee also found confusing the wording about edge roughness -- it seems at the first glance that TEM confirms that sample 2 has more edge roughness than Sample 1. In fact, TEM only tells that appearing edge roughness in method, equivalent to the one used for Sample 1, is about 1nm. It does not give any information about Sample 2.

This may need to be cleared as well.

3. Theoretical part of the work may require some extra cleaning, taking into account comment #1. Also, it might be good to understand if these result do go beyond Ref.26 or not. If not original, Referee would recommend moving more of this into Supplementary.

In conclusion, the paper may be published after revisions.

Reviewer #2 (Remarks to the Author):

The quantization of conductance in a quantum point contact (QPC) and in narrow quasi-dimensional wires represents a cornerstone of mesoscopic physics. The conductance quantization of QPCs defined in two-dimensional electron gas (2DEG) in semiconductor heterostructures was discovered in early eighties and then was well-explained within a one-electron picture of non-interacting electrons. This discovery generated a tremendous interest to various aspects of conductance quantization, including quantum Hall physics and the role of electron interaction. In the present article, based on earlier theoretical predictions, the authors demonstrate that the conductance quantization in graphene constrictions are strikingly different from the commonly accepted picture that holds in conventional 2DEG semiconductor heterostructures. The suppression of the conductance quantization in graphene is attributed to the electron interaction that changes qualitatively predictions based on the non-interacting picture. The experimental results reported in this paper as well as analysis of the experimental data are convincing and are backed up by solid theoretical calculations. Thus, because of fundamental significance of the reported results I recommend this article for publication.

Several comments and suggestions:

1) The suppression of the conductance quantization in the system at hand was attributed to the

effect of the electron interaction. The role of the electron interaction for the suppression of the conductance quantization in QPCs for the first time was outlined by the Cambridge group (Thomas et al., Phys. Rev. Lett., 77, 135 (1996)) who discovered the famous "0.7 anomaly" in the conductance in QPC. The experimental discovery of the "0.7 anomaly" has generated a tremendous experimental and theoretical interest to the role of the electron interaction in the quantization of QPC conductance. I suggest to mention the activity related to the "0.7 anomaly" in the introduction, which will further signify the fundamental importance of experimental findings reported in the present study.

1a) Another manifestation of the electron interaction in QPC and related systems is a so-called "0.25 feature" in the nonequilibrium differential conductance (see, e.g. Chen et al., Appl. Phys. Lett. 93, 032102 (2008)) whose detailed origin is still under debate.

2) On page 7 the authors write, "... Although demonstrated here in graphene, this is an universal phenomenon[4,9-11,25] occurring in ballistic, narrow conducting systems exhibiting hard-wall potential confinement such as semiconducting and metallic 2D crystals. ...". A conceptual similar physics is predicted to take place also in cleaved-edge overgrown quantum wires in the integer quantum Hall regime (Ihnatsenka et al., Phys. Rev. B 74, 075320). The cleaved-edge overgrown quantum wires, like graphene, are also characterized the hard-wall confinement and therefore shows similar band structure and the structure of the edge state.

Reviewer #3 (Remarks to the Author):

In this work, the authors study the effect of the edge quality in etched graphene nanoconstrictions on the conductance quantization in the quantum Hall regime. By using two different etch recipes, they produce constrictions with different edge roughness, which is shown to have an effect on the conductance of their devices at zero and positive magnetic fields.

In my view, the manuscript is interesting and well written, although a few points should still be clarified for a broader audience as I explain below. I believe the present work could possibly be accepted for publication after revisions.

In order to help to improve the manuscript and clarify a few points, I have a few questions and comments:

1) The authors use the non-uniform carrier density due to electrostatics and electron(-electron) interactions almost interchangeably throughout the main text and supplementary information. These are very distinct effects, and even though both effects might take place and contribute to the lack of conductance quantization at high magnetic fields in their devices, I believe that the authors should pin-point the main cause for this effect. For example, increasing the gate oxide thickness should increase the homogeneity of induced charge carriers in their devices. The text should also be revised to make clear that these two effects are distinct and could both contribute to their results.

2) I believe that once question 1 is addressed properly, the authors should put their work into perspective, not only addressing the reasons why other groups did not observe such effect but also as an indication of how important electron-electron interactions are in relation to geometrical effects.

3) Even though the authors state that the geometrical factors for the quantum Hall effect (R_{xx} to R_{xy} contribution due to the sample dimensions) cannot explain their results, I suggest that they explain their reasoning in a little more detail, perhaps calculating the R_{xx} and R_{xy} contribution to the two probe conductance for their samples or performing a fit as done in reference 26.

4) The constrictions show weak signs of conductance quantization at $B=0$ T, indicating quasi-ballistic transport. It would be relevant to use the semi-classical relation $G = (4e^2/h) * (k_F * W/\pi)$ to obtain an "effective width" of the constriction.

5) What is the magnetic field necessary to suppress the plateaus in conductance? Does it correspond to the expected fields from the sample geometry?

6) The authors state that they observe more "noise" in the conductance signals from their Type 2 samples. I believe that what they observe is not noise, but Universal Conductance Fluctuations (UCF), which should appear when the phase-coherence length is comparable to the device dimensions. Furthermore, the fluctuations for type 1 and type 2 devices are approximately the same size ($\sim 0.5 e^2/h$) even though the authors claim they are larger for the type 2 devices. Another indication of the presence of UCF is the absence of the fluctuations at high fields. This should be corrected throughout the manuscript and supplementary information.

Response to Reviewers' comments:

Reviewer #1 (Remarks to the Author):

REV1: The manuscript "Conductance quantization suppression in the quantum Hall regime" by Bøggild et.al is on an interesting topic of interplay between space and B-field quantization in 2D materials.

The manuscript seems to be free from obvious errors, and can be published with suggested minor revisions, as below.

Our response:

We thank the reviewer for judging our work being suitable for Nature Communications. We have answered to his/her comments as described below and modified the manuscript and supplementary material accordingly.

REV1: 1. Perhaps main reservation Referee had regarding Authors interpretation of term "electron interactions". Although broadly defined, this term has specific meaning of Coulomb e-e correlation, especially when using in contrast to "non-interacting electrons".

What is described in the paper is much simpler effect (not necessarily this is a drawback) -- the electrostatic screening of the gate potential by non-uniform distribution of charge carriers, resulting from both E- and B-fields *and* involving disorder of the edge.

Current manuscript is a big mess in this sense. Word "interaction" is overused and makes unprepared reader completely confused.

Our response:

We thank the Referee for stressing the importance of using precise terminology when addressing "electron interactions".

In our article, the inhomogeneous electrostatic potential across the device is calculated by combining the solution of the Poisson equation with the Thomas-Fermi approach as described in literature (Ref.[10]). This potential corresponds closely to those generated by more complex models, such as self-consistent solutions within the Hartree approximation (Refs. [9,29]). Although such approaches can account for both Coulomb interactions between injected carriers and their coupling to the external gate, as pointed out by the referee, the screening effect of the gate potential is the primary factor determining the charge density distribution in these systems (Ref.[30]).

We have made this argument clearer in the modified version the manuscript. In general, we have replaced the term “electron interactions” by “electrostatic interactions”, “Coulomb interactions” or “screening mechanism” where appropriate, and “non-interacting electrons” by “single-electron” or “one-electron”.

We have expanded the introductory part by commenting on other archetypical studies about electron-electron interactions (Refs. [12,13]). Further, we have emphasized that our study concerns the electrostatic screening of the gate potential in narrow, gated graphene devices. By doing so, we specify and single out the importance of our experimental findings. These changes are in line with the suggestions from another referee (Referee #2), too.

In particular, we have added:

(Main text, page 2, line 13). “For example, the conductance of a 1D channel with repulsive electron-electron interactions vanishes in the presence of any scattering potential at $B = 0$ T.³ Furthermore, the observation of the “0.7 anomaly”¹² or the “0.25” feature¹³ in the conductance quantization of QPCs at $B = 0$ T, are also signatures of electron-electron interactions. In a perpendicular B , ...”

(Main text, page 2, line 23). “We focus on the ballistic conductance of gated quasi-1D systems, where screening theories predict conductance quantization suppression (CQS) in the QH regime ”

(Main text, page 3, line 3). “Although both interactions between carriers and their coupling to the external gate can affect conductance^{6,9}, it is the electrostatic screening of the gate potential which is the main contributing mechanism^{4,5,10} to the CQS effect.”

(Main text, page 8, line 10). “We have demonstrated the suppression of conductance quantization in the QH regime due to electrostatic interactions in gated graphene nanoconstrictions with low edge roughness.”

REV1: 2. Referee also found confusing the wording about edge roughness -- it seems at the first glance that TEM confirms that sample 2 has more edge roughness than Sample 1. In fact, TEM only tells that appearing edge roughness in method, equivalent to the one used for Sample 1, is about 1nm. It does not give any information about Sample 2.

This may need to be cleared as well.

Our response:

We have made clearer the fact that both types of samples used in the study have a certain degree of edge roughness. However, Sample type 1 has a much lower disorder than Sample type 2. This can be seen from the much higher conductance values at zero magnetic field of Sample type 1, in line with other studies in literature [Ref. 26].

We have indeed only assessed by TEM the roughness of Sample type 1 to be $<1\text{nm}$. This value is needed in our calculations to verify the persistence of the CQS effect with this level of roughness.

The following sentences in the revised manuscript clarify this issue:

(Main text, Page 5, line 20). “While all our devices have a certain degree of edge disorder, Sample type 1 (Figure 2a) was etched using reactive ion etching, which is known to produce less edge disorder than the oxygen plasma ashing²⁶ technique used for Sample type 2 (Figure 2b). Specifically, we achieve edge roughness below 1 nm in Sample type 1 (Supplementary Information), comparable to values obtained in nanoribbons with extremely low edge roughness fabricated by unzipping carbon nanotubes²⁰. ”

(Main text, Page 6, line 7). “However, conductance values in Sample type 1 are three times larger than those for Sample type 2. This specific behavior has previously been attributed to differences in edge disorder²⁶”

REV1: 3. Theoretical part of the work may require some extra cleaning, taking into account comment #1.

Our response:

Taking into account referee’s comment #1 and our answer to it, we have added the following sentence in the main text, when describing the used theory.

(Main text, page 7, line 11). “tight-binding calculations, where the inhomogeneous electrostatic potential across the gated device is introduced using the analytical model proposed by Silvestrov and Efetov¹⁰ (Methods). This potential corresponds closely to those generated by more complex models, such as self-consistent solutions within the Hartree approximation^{9,27,29}. Although such approaches can account for both Coulomb interactions between injected carriers and the

coupling to the external gate, the screening effect due to the latter is the primarily factor determining the charge density distribution in these systems³⁰”

REV1: Also, it might be good to understand if these results do go beyond Ref.26 or not. If not original, Referee would recommend moving more of this into Supplementary.

In conclusion, the paper may be published after revisions.

Our response:

Our theory results (Figure 3) go substantially beyond Ref. 26 (Ref. 28 in the new version of the manuscript). That reference calculates the two-terminal conductance of a homogeneous 2D conductor. Our theory calculates the conductance of an inhomogeneous conductor (as predicted in Refs.9,10), also accounting for the different degrees of edge roughness existing in experimental devices.

To further emphasize the differences between our used theory and that one of Ref. 26, and, as suggested by another referee (Referee #3), we have compared our experimental results with those from Ref. 26.

This comparison is shown in Supplementary Note 4 and Supplementary Fig. 8.

REV2: Reviewer #2 (Remarks to the Author):

The quantization of conductance in a quantum point contact (QPC) and in narrow quasi-dimensional wires represents a cornerstone of mesoscopic physics. The conductance quantization of QPCs defined in two-dimensional electron gas (2DEG) in semiconductor heterostructures was discovered in early eighties and then was well-explained within a one-electron picture of non-interacting electrons. This discovery generated a tremendous interest to various aspects of conductance quantization, including quantum Hall physics and the role of electron interaction. In the present article, based on earlier theoretical predictions, the authors demonstrate that the conductance quantization in graphene constrictions are strikingly different from the commonly accepted picture that holds in conventional 2DEG

semiconductor heterostructures. The suppression of the conductance quantization in graphene is attributed to the electron interaction that changes qualitatively predictions based on the non-interacting picture. The experimental results reported in this paper as well as analysis of the experimental data are convincing and are backed up by solid theoretical calculations. Thus, because of fundamental significance of the reported results I recommend this article for publication.

Our response:

We thank the reviewer for recommending our work for Nature Communications. We have addressed his/her comments to the manuscript as described below.

REV 2: Several comments and suggestions:

1) The suppression of the conductance quantization in the system at hand was attributed to the effect of the electron interaction. The role of the electron interaction for the suppression of the conductance quantization in QPCs for the first time was outlined by the Cambridge group (Thomas et al., Phys. Rev. Lett., 77, 135 (1996)) who discovered the famous “0.7 anomaly” in the conductance in QPC. The experimental discovery of the “0.7 anomaly” has generated a tremendous experimental and theoretical interest to the role of the electron interaction in the quantization of QPC conductance. I suggest to mention the activity related to the “0.7 anomaly” in the introduction, which will further signify the fundamental importance of experimental findings reported in the present study.

1a) Another manifestation of the electron interaction in QPC and related systems is a so-called “0.25 feature” in the nonequilibrium differential conductance (see, e.g. Chen et al., Appl. Phys. Lett. 93, 032102 (2008)) whose detailed origin is still under debate.

Our response:

We have included the suggested studies about electron-electron interactions (Refs.[12,13]). The new version of the manuscript reads:

(Main text, page 2, line 13). “For example, the conductance of a 1D channel with repulsive electron-electron interactions vanishes in the presence of any scattering potential at $B = 0$ T.³ Furthermore, the observation of the “0.7 anomaly”¹² or the “0.25” feature¹³ in the conductance quantization of QPCs at $B = 0$ T, are also signatures of electron-electron interactions. In a perpendicular B , ...”

REV2: 2) On page 7 the authors write, “... Although demonstrated here in graphene, this is an universal phenomenon^[4,9-11,25] occurring in ballistic, narrow conducting systems exhibiting hard-wall potential confinement such as semiconducting and metallic 2D crystals. ...”. A conceptual similar physics is predicted to take place also in cleaved-edge overgrown quantum wires in the integer quantum Hall regime (Ihnatsenka et al., Phys. Rev. B 74, 075320). The cleaved-edge overgrown quantum wires, like graphene, are also characterized the hard-wall confinement and therefore shows similar band structure and the structure of the edge state.

Our response:

We thank the reviewer for calling this classic system to our attention. We have mentioned that similar physics is predicted in cleaved-edge overgrown wires.

We have added:

(Main text, page 8, line 11). “Although demonstrated here in graphene, this is an universal phenomenon^{4,9-11,27} occurring in ballistic, narrow conducting systems exhibiting hard-wall potential confinement such as semiconducting and metallic 2D crystals or cleaved-edged overgrown quantum wires³¹”.

REV3: Reviewer #3 (Remarks to the Author):

In this work, the authors study the effect of the edge quality in etched graphene nanoconstrictions on the conductance quantization in the quantum Hall regime. By using two different etch recipes, they produce constrictions with different edge roughness, which is shown to have an effect on the conductance of their devices at zero and positive magnetic fields.

In my view, the manuscript is interesting and well written, although a few points should still be clarified for a broader audience as I explain below. I believe the present work could possibly be accepted for publication after revisions.

Our response:

We thank the reviewer for judging our work to be interesting for Nature Communications. We have answered to his/her comments as described below and modified the manuscript and supplementary material accordingly.

REV3: In order to help to improve the manuscript and clarify a few points, I have a few questions and comments:

1) The authors use the non-uniform carrier density due to electrostatics and electron(-electron) interactions almost interchangeably throughout the main text and supplementary information. These are very distinct effects, and even though both effects might take place and contribute to the lack of conductance quantization at high magnetic fields in their devices, I believe that the authors should pin-point the main cause for this effect. For example, increasing the gate oxide thickness should increase the homogeneity of induced charge carriers in their devices. The text should also be revised to make clear that these two effects are distinct and could both contribute to their results.

Our response:

We thank the Referee for pertinent comments. In the revised manuscript we try to be clearer in the introduction, stating that our effects are mainly due to screening effects due to the gate potential rather than electron-electron interactions. We have avoided the usage of the ambiguous term “electron interactions”.

This is also now clarified in the theoretical description, where we specify that the screening mechanisms due to the gate are the primary factor determining the inhomogeneous charge carrier distribution and thus the CQS effect.

In particular, we have written/inserted:

(Main text, page 2, line 23). “We focus on the ballistic conductance of gated quasi-1D systems, where screening theories predict conductance quantization suppression (CQS) in the QH regime ”

(Main text, page 3, line 3). “Although both interactions between carriers and their coupling to the external gate can effect conductance^{6,9}, it is the electrostatic screening of the gate potential which is the main contributing mechanism^{4,5,10} to the CQS effect.”

(Main text, page 7, line 11). **“tight-binding calculations, where the inhomogeneous electrostatic potential across the gated device is introduced using the analytical model proposed by Silvestrov and Efetov¹⁰ (Methods). This potential corresponds closely to those generated by more complex models, such as self-consistent solutions within the Hartree approximation^{9,27,29}. Although such approaches can account for both Coulomb interactions between injected carriers and the coupling to the external gate, the screening effect due to the latter is the primary factor determining the charge density distribution in these systems³⁰”**

REV3: 2) I believe that once question 1 is addressed properly, the authors should put their work into perspective, not only addressing the reasons why other groups did not observe such effect but also as an indication of how important electron-electron interactions are in relation to geometrical effects.

Our response:

Thank you for this suggestion to help place our work in context. We have expanded our discussion on the importance of device geometries (sample dimensions and dielectric thicknesses) in enhancing electrostatic (screening) effects.

In particular, we have added:

(Main text, page 3, line 10). **“Graphene, on the other hand provides extraordinary opportunities to examine the physics of the QH effect¹⁶. First, it exhibits a natural hard-wall confinement at its borders. Furthermore, the distance to the gate can be arbitrarily selected since electrons in strict 2D materials reside right at the surface. Both features enable the possibility of designing specific device geometries which are (heavily) dominated by screening effects. An example of such a geometry is a narrow graphene strip with a width comparable or smaller than the thickness of the dielectric spacer^{9,10}.”**

(Main text, page 4, line 2). **“specifically, a device geometry able to produce a large charge density gradient across the nanostructure,...”**

REV3: 3) Even though the authors state that the geometrical factors for the quantum Hall effect (R_{xx} to R_{xy} contribution due to the sample dimensions) cannot explain their results, I suggest that they explain their reasoning in a little more detail, perhaps calculating the R_{xx}

and Rxy contribution to the two probe conductance for their samples or performing a fit as done in reference 26.

Our response:

We have included a section in the supplement (Supplementary Note 4 and Supplementary Fig. 8), explaining in detail how geometrical factors in homogeneous two-terminal devices cannot account for the measured magnetoconductance data in our nanoconstrictions with smooth edge disorder where the CQS effect is seen (Sample Type 1).

In contrast, as we show in that supplementary section, the magnetoconductance data of our nanoconstrictions with rougher edges (“Sample type 2”) and our measured graphene strips of width $W=1\mu\text{m}$ do indeed agree with these geometrical corrections, in support of notion of the latter two geometries possessing a homogeneous charge density across the device.

REV3: 4) The constrictions show weak signs of conductance quantization at $B=0$ T, indicating quasi-ballistic transport. It would be relevant to use the semi-classical relation $G = (4e^2/h) \cdot (k_F \cdot W/\pi)$ to obtain an “effective width” of the constriction.

Our response:

We thank the reviewer for suggesting us to do this analysis, as we find that this is a rather clear visualization of the difference between the two sample types, in terms that are well-known to a broader audience.

We have added this information into a corresponding paragraph within the Supplementary Note 3.

In order to compare further the quality of both types of etched nanoconstrictions, we have estimated the effective width of our constrictions by taking into account the semi-classical relation $G = (4e^2/h) \cdot (k_F \cdot T \cdot W_{\text{phys}}/\pi)$ Refs [17,18]. In the former expression the effective width W requested by the Referee is given by $W = T \cdot W_{\text{phys}}$, where W_{phys} is the physical width of the constrictions (100 nm in our case) . We explicitly write the average transmission parameter $0 < T < 1$, in order to relate it directly to the different degrees of edge roughness in our constrictions.

For this calculation, we take $k_F = \sqrt{\pi \cdot n_0}$, where n_0 is the charge density calculated for an infinite capacitor. We note that this is an approximation since the averaged charge density n_{avg} of nanostructures with inhomogeneous charge density $n(x)$ is larger than n_0 . In particular, in the case of graphene strips, $n_{\text{avg}} = n_0 \cdot \pi/2$ Ref.[10].

The newly added Supplementary Figure 6 shows the averaged transmission coefficient of our devices, comparing it with the case of perfect transmission $T = 1$ for a constriction of width $W_{\text{phys}} = 100$ nm. Within our approximation, we can show that nanoconstrictions of Sample type 1

(smooth edges) have a transmission coefficient larger than 0.7, meanwhile nanoconstrictions of Sample type 2 (rougher edges) have a transmission coefficient below 0.3, confirming the fact that edge roughness plays a key role in the charge transport of narrow devices.

REV3: 5) What is the magnetic field necessary to suppress the plateaus in conductance? Does it correspond to the expected fields from the sample geometry?

Our response:

In a two-dimensional conductor subjected to screening effects, the plateaus in conductance in the quantum Hall regime are suppressed if the device is free of stable incompressible strips (thus, the width of these strips is narrower than the magnetic length). This indicates that plateaus in the conductance for the given nanostructure are suppressed even at low magnetic fields until a critical field is reached where the following condition (Eq. 3, main text) is no longer valid:

$$\frac{dn_{el}(x)}{dx} \geq \frac{\varepsilon v_F}{\pi^2} \left(\frac{k}{\hbar e} \right)^{1/2} (2B)^{3/2}, \text{ (Eq. R1)}$$

In the case of narrow graphene devices (ribbons), we can further estimate the inhomogeneous charge density needed across a device $n(x)$ for the CQS effect to appear using an edge-state approach. The minimum charge density across the nanostructure needed to introduce at least one pair of counter-propagating edge-states in the central part of the device is (Ref. [10], Supplementary Note 6):

$$\frac{n(x)}{n_0} > \frac{2e^2 b B}{\hbar \pi \varepsilon V_g} \text{ (Eq. R2)}$$

Similar to Eq. R1, this equation (Eq. R2) indicates that the increased conductance with suppressed plateaus in a given nanoribbon occurs from low B fields. This is valid when increasing B until a critical field is reached where the aforementioned condition is no longer valid. The exact magnetic field at which the quantization is restored will highly depend on the sample quality (in particular gate voltages at which Landau levels are reached for a given B or edge roughness effects).

Experimentally, we can confirm that the CQS phenomenon exists at lower fields, too. Plateaus in the quantum Hall regime are clearly seen for our graphene quality (graphene strips with width $W = 1 \mu\text{m}$) for $B=3\text{T}$ (see Supplementary Fig. 2). At $B=3-4\text{T}$, the CQS phenomenon in our narrow graphene constrictions is already seen in the LL0 (see Supplementary Fig. S15), too.

Following the suggestion of the referee, we have included this discussion in the Supplementary Note 6.

REV3: 6) The authors state that they observe more “noise” in the conductance signals from their type 2 samples. I believe that what they observe is not noise, but Universal Conductance Fluctuations (UCF), which should appear when the phase-coherence length is comparable to the device dimensions. Furthermore, the fluctuations for type 1 and type 2 devices are approximately the same size ($\sim 0.5 e^2/h$) even though the authors claim they are larger for the type 2 devices. Another indication of the presence of UCF is the absence of the fluctuations at high fields. This should be corrected throughout the manuscript and supplementary information.

Our response:

We thank the Referee for underlining the use of proper terminology. We meant “noisier” measurements in terms of the presence of more rapidly varying fluctuations with respect to the gate voltage, not in terms of the conductance magnitude. As suggested by the Referee, we have eliminated the word “noise” thorough the manuscript to avoid confusion.

Furthermore, we have examined the magnitude of the conductance fluctuations ($G(T=300\text{K})-G(T=4\text{K})$) and confirm that they are $\sim 0.5 e^2/h$ in both type of constrictions, as pointed out by the Referee.

We have inserted the following sentence in the Supplementary Note 3:

“Conductance “kinks” in Sample type 2 are less obvious due to the presence of more rapidly varying fluctuations in their electrical characteristics with respect to the gate voltage^{S12}. In addition, we note that conductance fluctuations in both types of samples have an approximate magnitude of $\sim 0.5e^2/h$, indicative of Universal Conductance Fluctuations (UCF) Ref.[S16]. This reasoning is corroborated by the absence of such fluctuations at high magnetic fields.”

REVIEWERS' COMMENTS:

Reviewer #1 (Remarks to the Author):

Authors adequately addressed Referee's questions. Paper can be published.

Reviewer #2 (Confidential remarks to the Editor).

Reviewer #3 (Remarks to the Author):

In their revised version of the manuscript the authors address all the questions and concerns raised by me and the other referees by extensively rewriting and clarifying several points in the main text and adding extra information and analysis in the supplementary information to support their claims.

My main concern about the previous version of the manuscript was that the text was unclear if the main contribution for the effect the authors observe was coming from electron-electron interactions or electrostatics. This has been clarified in their current version where they explicitly mention that electrostatics and screening are the important mechanisms for the conductance quantization suppression in their devices. This is also put in perspective with other works where this effect has not been observed which, according to the authors, is a combination of sample geometry, quality and dimensions. In my opinion, this is a valid claim.

The authors also thoroughly answer the remaining points I raised in my last report with additions to the supplementary information text.

I believe the work presented here shows an intriguing effect which has enough potential to be attractive to the 2D materials, 2DEG, and the general magneto-transport communities. Therefore, in my opinion, the manuscript can be accepted for publication in Nature Communications.

AUTHORS RESPONSES

REV 1. Reviewer #1 (Remarks to the Author):

Authors adequately addressed Referee's questions. Paper can be published.

Our response:

We thank the reviewer for recommending our work for Nature Communications.

REV.3 Reviewer #3 (Remarks to the Author):

In their revised version of the manuscript the authors address all the questions and concerns raised by me and the other referees by extensively rewriting and clarifying several points in the main text and adding extra information and analysis in the supplementary information to support their claims. My main concern about the previous version of the manuscript was that the text was unclear if the main contribution for the effect the authors observe was coming from electron-electron interactions or electrostatics. This has been clarified in their current version where they explicitly mention that electrostatics and screening are the important mechanisms for the conductance quantization suppression in their devices. This is also put in perspective with other works where this effect has not been observed which, according to the authors, is a combination of sample geometry, quality and dimensions. In my opinion, this is a valid claim.

The authors also thoroughly answer the remaining points I raised in my last report with additions to the supplementary information text.

I believe the work presented here shows an intriguing effect which has enough potential to be attractive to the 2D materials, 2DEG, and the general magneto-transport communities. Therefore, in my opinion, the manuscript can be accepted for publication in Nature Communications.

Our response:

We thank the reviewer for recommending our work for Nature Communications.